# Role of Citicoline in the Management of Traumatic Brain Injury

**DOI:** 10.3390/ph14050410

**Published:** 2021-04-26

**Authors:** Julio J. Secades

**Affiliations:** Medical Department, Ferrer, 08029 Barcelona, Spain; jsecades@ferrer.com

**Keywords:** CDP-choline, citicoline, pharmacological neuroprotection, brain ischemia, traumatic brain injury, head injury

## Abstract

Head injury is among the most devastating types of injury, specifically called Traumatic Brain Injury (TBI). There is a need to diminish the morbidity related with TBI and to improve the outcome of patients suffering TBI. Among the improvements in the treatment of TBI, neuroprotection is one of the upcoming improvements. Citicoline has been used in the management of brain ischemia related disorders, such as TBI. Citicoline has biochemical, pharmacological, and pharmacokinetic characteristics that make it a potentially useful neuroprotective drug for the management of TBI. A short review of these characteristics is included in this paper. Moreover, a narrative review of almost all the published or communicated studies performed with this drug in the management of patients with head injury is included. Based on the results obtained in these clinical studies, it is possible to conclude that citicoline is able to accelerate the recovery of consciousness and to improve the outcome of this kind of patient, with an excellent safety profile. Thus, citicoline could have a potential role in the management of TBI.

## 1. Introduction

Traumatic brain injury (TBI) is among the most devastating types of injury and can result in a different profile of neurological and cognitive deficits, and even death in the most severe cases. TBI represents a large portion of the global injury burden and is caused mainly by falls, especially in old patients, and road injuries, often affecting young patients [1]. The effects of TBI are not limited to the patient suffering from this injury; it also affects families and societies, with a relevant financial burden. There is a solid agreement that the management of TBI must be focused to avoid brain injury. Upon clinical examination [2], TBI is classified into mild, moderate, and severe, based on the scores of the Glasgow Coma Scale (GCS). Such categories have been found to be predictive of a patient’s long-term outcome, although other measures and models have also been tested, as biomarkers [2,3].

As explained, the management of TBI has to be focused on the reduction of the severity of the sequelae and to improve the recovery of these. The improvement in monitoring and in the knowledge of the pathophysiology of TBI could change current management, allowing for more adequate interventions that could improve the final outcomes, reducing the associated disabilities [4]. This improvement has been based on a better understanding of the complex pathophysiology of TBI. Neuroprotection is considered one of the treatments among the improvements in the treatment of TBI [5,6,7,8,9].

Among the biochemical mechanisms implicated in the pathophysiology of TBI, the inflammatory processes [10,11,12,13,14,15,16], and the impairment of the phospholipid metabolism and its consequences [17,18,19,20,21,22,23,24,25,26,27,28,29,30,31,32,33,34,35] play an important role. Because of these pathophysiological conditions, there is agreement on the need for drugs that may have a protective and restorative or reparative effect on the nervous system [36,37,38,39,40], the so-called “traumatic penumbra” being the target of their effect [22]. Citicoline is the generic name for cytidine-5′-diphosphocholine or CDP-choline, which is a normal component of the human metabolism as an intermediate in the synthesis of phosphatidylcholine, the main phospholipid in cellular membranes (Figure 1), and, when it is administered exogenously as a drug, has a wide range of biochemical and pharmacological actions becoming a putative treatment for some neurological diseases [41], including TBI [22].

## 2. Experimental Data

Table 1 summarizes the most relevant preclinical studies evaluating the therapeutic efficacy and the mechanism of action of citicoline in different experimental models.

It has been demonstrated that citicoline is able to prevent degradation of choline and ethanolamine phospholipids during brain ischemia [42], and to restore the integrity of the blood−brain barrier [43]. Other old experimental studies have shown the protective effects of citicoline in neuronal cultures in hypocapnic conditions [44], and positive effects on the threshold for the arousal reaction, reducing the duration of coma induced by different mechanisms [45,46,47,48]. For example, Ogashiwa et al. [48] analyzed the dose-response of citicoline on post-traumatic disturbances of consciousness in mice. They used groups of 10 mice for every dose of citicoline administered by IV. The dose-escalation of citicoline was 0, 1, 2, 4, 8, 15, 30, 60, 125, and 250 mg/Kg and the assignment was randomized. Control group received comparable volumes of saline solution. The injury (mechanical impact) was induced 10 min after the administration of the drug. The degree of concussion was assessed measuring the time interval from the onset of coma and the return of the righting reflex (RR time) and the time between the onset of coma and subsequent appearance of spontaneous movement (SM time). A significant dose-dependent relationship was found on RR time in animals that received the doses of 60 mg/Kg or higher. Furthermore, for the same level of doses, a significant reduction of SM times was found, thus the authors concluded that with doses not less than 60 mg/kg IV, citicoline showed a statistically significant improvement in the disturbances of consciousness during the acute phase of head injury in mice.

Citicoline can increase the incorporation and the metabolism of glucose and reduce the levels of lactate in the brain during ischemia [49]. Citicoline can modulate the activity of some enzymes, such as cholinephosphotransferase or phospholipases, especially phospholipase A2 [50,51,52,53,54,55]. Arrigoni et al. [51] demonstrated that following a cryogenic injury of the brain, within the tissue surrounding the necrotic lesion, there was a clear activation of the phospholipase A2 at 2 and 4 h after the lesion that participated in the phospholipid breakdown. They also detected an activation of the cholinephosphotransferase 2 h post-injury, that could correspond to an early phospholipid resynthesis. For their experiment, the authors used cryogenic brain injury in rabbits that were randomly assigned to the control group or to the lesion group. Animals were killed by decapitation 2, 4 and 8 h after the lesion. Eighteen injured animals received 200 mg/Kg PO of citicoline 5 h before the injury, except for the 8 h-injured group which received 100 mg/Kg PO 5 h before and 1 h after the injury. Eighteen injured animals were used as controls. The enzymatic activities were measured in the brain homogenate obtained after the sacrifice. As described, a significant increase of the activity of the phospholipase A2 was detected 2 and 4 h after the injury. In injured animals, citicoline administration was associated with a significant prevention of the phospholipase A2 stimulation as compared to the untreated injured animals, whereas the drug did not affect the phospholipase A2 activity in control treated animals. In this model, citicoline did not affect the cholinephosphotransferase activity in any experimental conditions. The authors concluded that the beneficial effects of citicoline might be explained by a prevention of destruction rather than an enhancement of reconstruction of phospholipids. Kitazaki et al. [54] studied the effects of citicoline on membrane-associated phospholipase A2 using a synaptosome fraction from rat cerebral cortex and a crude membrane fraction from rat platelets. Citicoline, at a dose 1–10 mM, inhibited the activity of both enzymes I a dose-dependent manner, being this inhibition competitive.

In some experimental models, citicoline provided significant protection against the lethality, improving the survival quality [56,57,58]. Algate et al. [56] compared the effects of repeated oral administration of citicoline and vehicle on the EEG changes induced by epidural brain compression in anesthetized cats. Animals were treated with citicoline at doses 0.5 or 2.5 g/Kg per day for 5 days. After this 5 days of treatment, animals were anesthetized and the epidural compression was induced by a balloon catheter and platinum electrodes were placed stereotactically to record EEG. Citicoline-treated cats exhibited a statistically significant increase in resistance to the effects of mechanical compression of the brain when compared with the vehicle-treated group. The onset of abnormal EEG waveforms occurred at greater levels of brain compression in citicoline-treated animals, and these animals were less susceptible to cardiac and respiratory disorders. A statistically significant protection against the lethal effects of epidural compression was also noted in treated animals. The authors concluded that repeated oral treatment with citicoline can provide some protection against the effects of acute mechanical compression of the brain. Kondo [58] studied the effects of citicoline (15–20 mg/Kg) slowly injected into the common carotid artery or IV in a model of progressive brain compression in cats. Citicoline markedly improved the disturbances of consciousness and the EEG records. It has been possible to detect the labelled drug in the brain, notably in the affected areas in a model of cold brain injury in rats [59]. Boismare et al. [60] studied the effects of citicoline in a model of craniocervical trauma without direct blow (“whiplash”) in rats. The lesion was induced by means of a fierce acceleration and deceleration applied to the animals. Two days after the lesion animals presented an asymmetry in the activity of the latero-cervical muscles, a postural dysregulation of the brain circulation, and disturbances in conditioned avoidance responses. These alterations were related with an increase of the cerebral norepinephrine splitting. The intraperitoneal injection of citicoline (20 mg/Kg) one hour after the trauma prevented all the modifications on the central catecholamine metabolism and suppressed the behavioral disorders noticed in the control animals. In many different experimental models of brain edema a significant effect of citicoline on the reduction of the edema has been demonstrated, associating this effect with the restoration of the activity of the membranous ATPases that facilitate the reabsorption of the edema [17,18,61,62,63,64,65,66]. Clendenon et al. [61] studied the effects of citicoline (100–200 mg/Kg IV) on the ATPase activity in a model of spinal cord injury by impact in dogs. Two minutes after the injury a significant decrease of the Na+/K+-ATPase activity was found, and at 30 min, also decreased activity of the Mg++-ATPase. Administration of citicoline 5 min after injury prevented the decrease of the Mg++-ATPase activity, whereas the decrease of the Na+/K+-ATPase activity was not prevented, probably because it was already inactivated to a low level before the administration of the drug. Cohadon et al. [62] demonstrated that in the development of vasogenic cerebral edema there was a progressive quantitative and qualitative impairment of mitochondrial-ATPase and of Na+/K+-ATPase using a model of cryogenic brain edema in rats. Citicoline was administered 24 h after the injury at doses of 20 mg/Kg and continued daily until the sacrifice of the animals. Citicoline was able to correct this disturbed enzymatic activity and at the same time reduce the extent of the cerebral edema. Cervós-Navarro and Lafuente [63,64] studied the effects of citicoline in a model of ultraviolet-induced brain edema in cats. After the induction of the edema by exposure to ultraviolet light, eight cats were treated with a single dose of citicoline of 100 mg/Kg 1, 8 and 16 h after the lesion. Eight cats were used as controls and they received the same volume of solvent alone (1 mL/Kg). Tissue water content was measured by a microgravimetric method. It was shown that animals treated with citicoline had a less volumetric increase than those animals treated with placebo, demonstrating an effect of the drug accelerating the reabsorption of brain edema. Majem et al. [65] analyzed the effects of multiple oral doses of citicoline on the bioelectric changes induced by brain edema in a model of cryogenic brain edema in rats. Treated animals (*n* = 8) receive oral citicoline at a dose 1 g/Kg/d for 8 days, starting immediately after the lesion. Treated animals showed a significant increase of the theta band during their awake state at the expense of a decrease of delta band as compared to control animals. A minor interindividual dispersion in each of the frequency bands was observed in treated animals, that somehow reflected the protective effects of the drug. Roda [66] in his study assessing the effects of citicoline on two different animal models of cryogenic brain edema, demonstrated the efficacy of the drug by means of a significant reduction of the Evans blue extravasation and that this effect was more evident when the drug was administered before the induction of the injury. Citicoline has been able to show a significant effect on microvascular permeability during experimental endotoxemia [67] and in models of early burn edema [68], postulating for a significant anti-inflammatory effect of the drug.

The effects of exogenous administration of citicoline (100 mg/Kg/d/18 d IP beginning 1 day postinjury) on the motor consequences, spatial memory capacity, and acetylcholine levels in some areas of the brain, such as hippocampus and neocortex were analyzed in an experimental model of traumatic brain injury by a controlled lateral impact in a total of 50 rats. In the motor study, animals treated with citicoline had a significantly longer balance period than animals in the control group. In addition, the treatment with citicoline was significantly associated with less cognitive deficits. This effect could be explained by the rapid increase in acetylcholine production seen after a single administration of citicoline, in microdialysis studies [69].

Plataras et al. [70] investigated the effect of different citicoline concentrations (0.1–1 mM) on acetylcholinesterase, Na+/K+-ATPase and Mg++-ATPase activities in homogenates of adult rat whole brain and in pure (nonmembrane bound) enzymes. After a 1–3 h citicoline preincubation, this drug induced a maximal stimulation of 20–25% (*p* < 0.001) for acetylcholinesterase and 50–55% (*p* < 0.001) for Na+/K+-ATPase, but it did not influence Mg++-ATPase activity. Citicoline can stimulate brain acetylcholinesterase and Na+/K+-ATPase independently of acetylcholine and noradrenaline and these effects could account for the clinical effects of the drug [70].

Başkaya et al. [71] examined the effect of citicoline on secondary injury factors, brain edema and blood−brain barrier breakdown, after TBI using a model of controlled cortical impact in rats. Brain edema was evaluated using the wet-dry method 24 h postinjury, and blood−brain barrier breakdown was evaluated by measuring Evans blue dye extravasation with fluorescein 6 h after TBI. After the injury, treated animals received intraperitoneal injections of citicoline (50, 100, or 400 mg/kg two times after TBI (*n* = 8–10 animals in each group)) and controls animals received saline solution (8 animals). TBI produced an increase in the water content and in measuring Evans blue dye extravasation in the injured cortex and the ipsilateral hippocampus. Citicoline at a dose of 50 mg/kg had no significant effect. At a dose of 100 mg/kg, citicoline attenuated Evans blue dye extravasation in both regions, although it reduced brain edema only in the injured cortex. In both regions, 400 mg/kg of citicoline significantly decreased brain edema and blood−brain barrier breakdown. Thus, the authors demonstrated a dose-dependent neuroprotective effect of citicoline in a model of experimental TBI, suggesting that this drug could be an effective neuroprotective agent on secondary injuries that appear following TBI.

Dempsey et al. [72] investigated whether citicoline protects the hippocampal neurons after controlled cortical impact-induced TBI in adult rats. Citicoline (100, 200, and 400 mg/Kg) or saline were injected intraperitoneally into the animals twice (immediately postinjury and 6 h postinjury). Seven days after the injury, the rats were neurologically evaluated and killed, and the number of hippocampal neurons was estimated by examining thionine-stained brain sections. By 7 days postinjury, there was a significant amount of neuronal death in the ipsilateral hippocampus in the CA2 (53 ± 7%, *p* < 0.05) and CA3 (59 ± 9%, *p* < 0.05) regions and a contusion (volume 34 ± 8 mm^3^) in the ipsilateral cortex compared with sham-operated control animals. Rats subjected to TBI also displayed severe neurological deficit at 7 days postinjury. Treating rats with citicoline at doses of 200 and 400 mg/Kg significantly prevented TBI-induced neuronal loss in the hippocampus, decreased cortical contusion volume, and improved neurological recovery.

Menku et al. [73] demonstrated a synergistic effect in the association of propofol with citicoline in an experimental model of TBI in rats, resulting in a higher reduction of the lipidic peroxidation when the drugs were administered together, concluding that this combination therapy may become a feasible option for the treatment of head injury.

Jacotte-Simancas et al. [74], using a model of controlled cortical impact injury in rats, studied the effects of citicoline and/or voluntary physical exercise on the related memory deficits and on neurogenesis and neuroprotection. Forty-eight male Sprague Dawley albino rats, six-weeks old, were randomly assigned to one of five experimental conditions, according to whether they were sham operated or had received TBI, and according to the treatment administered: citicoline vs. vehicle and exercise (E) vs. sedentary conditions. Citicoline (200 mg/kg) was administered by intraperitoneal route 4 h after surgery, and thereafter every 24 h until completing five injections. Citicoline improved memory deficits at short and long-term, while physical activity only in the long-term test. Physical activity increased cell proliferation and neurogenesis, and citicoline reduced the interhemispheric differences in the volume of the hippocampal formation. The combined effects of citicoline and physical exercise did not show any synergy, even the present data could suggest that the combined treatment with citicoline and physical exercise should be avoided in patients after TBI.

Qian et al. [75] designed an experimental study to investigate the neuroprotective effects of citicoline on a model of closed head injury in rats. Citicoline (250 mg/Kg IV 30 min and 4 h after injury) lessened body weight loss, and improved neurological functions significantly at 7 days. Treatment with citicoline was associated with a decrease of brain edema and of blood−brain barrier permeability, an enhancement of the activities of superoxide dismutase and the levels of glutathione, and with a reduction of the levels of malondialdehyde and lactic acid. Moreover, citicoline suppressed the activities of calpain, and enhanced the levels of calpastatin, myelin basic protein and αII-spectrin in traumatic tissue 24 h after injury. Citicoline was also able to attenuate the axonal and myelin sheath damage in corpus callosum and the neuronal cell death in hippocampal CA1 and CA3 subfields 7 days after injury. Authors consider that these findings provide additional support for the use of citicoline in the management of TBI.

Gan et al. [76], in an in vivo TBI zebrafish model, demonstrated that microglia, considered the resident macrophages of the central nervous system, accumulated rapidly after the injury. To perform its function, activated microglia secreted two types of cytokines, including proinflammatory interleukins and anti-inflammatory factors, helping to remove injured neurons and restore the homeostasis of the central nervous system. Citicoline was able to induce a further activation of microglia, and this was related with the reduction of neuronal apoptosis and the promotion of neuronal proliferation around the lesioned site associated with the use of citicoline.

Furthermore, some positive neuroprotective effects of citicoline have been published on different models of traumatic spinal cord lesion [77,78,79,80]. 

**Table 1 pharmaceuticals-14-00410-t001:** Summary of preclinical studies evaluating the therapeutic efficacy and the mechanism of action of citicoline in TBI.

Author	Year	Study Design	Experimental Model	Insult	Dose	Main Results
Tsuchida et al. [59]	1967	Comparative study	Rats	Cold injury	100µc ^3^H-CDP-choline IP	Significant incorporation of the labelled drug in the affected areas of the brain
Kondo [58]	1968	Comparative study	Male cats (2.5–4 Kg)	Epidural compression	15–20 mg/Kg intracarotid	Significant increase of survival rates
Boismare et al. [60]	1977	Comparative study	Rats	Whiplash injury	20 mg/Kg IP	Significant prevention on catecholamines changes in brain and suppression of behavioral disorders
Cohadon F. et al. [62]	1979	Comparative study	Rabbits (~2.5 Kg)	Cryogenic lesion	20 mg/Kg/4 d IV (starting 24 h after injury)	Significant restoration of the activity of the mitochondrial ATPase and of the membranous Na+/K+-ATPase. Acceleration of the reabsorption of brain edema.
Roda J.E. [66]	1980	Comparative study	Wistar Rats	Cryogenic lesion	6 mg/Kg/12 h IP Starting 24 h before the lesion and continued until sacrifice	Significant reduction of the extravasation of blue Evans
			Cats	Cryogenic lesion	15 mg/Kg/12 h IP Starting 24 h before the lesion or 2 h after the lesion and continued until sacrifice	Significant reduction of the extravasation of blue Evans. Better results when administered before the lesion
Algate et al. [56]	1983	Comparative study	Male cats (2.65–3.65 Kg)	Epidural compression	0.5 g/Kg/5 d PO	Significant increase in resistance to effects of mechanical compression
Ogashiwa M. et al. [48]	1985	Comparative randomized study	Mice	Mechanical impact	1–250 mg/Kg IV	Significant dose-effect on duration of coma
Kitazaki T. et al. [54]	1985	Comparative study	Rats	N/A ^a^	1–10 mM	Dose-dependent inhibition of activity of PLA2 ^b^
Clendenon et al. [61]	1985	Comparative study	Mongrel dogs (8–12 Kg)	Impact injury at spinal cord	100–200 mg/Kg IV	Prevention of the decrease of Mg^2+^-dependent ATPase activity
Lafuente J.V. et al. [63].	1986	Comparative randomized study	Male cats	Ultraviolet-induced brain edema	20 mg/Kg IV	Significant acceleration of the reabsorption of brain edema
Majem X. et al. [65]	1986	Comparative study	Male rats (180–200 g)	Cryogenic lesion	1 g/Kg/8 d PO	Significant increase of theta activity and decrease of delta activity on EEG
Arrigoni E. et al. [51]	1987	Comparative randomized study	Female rabbits (2.0–2.5 Kg)	Cryogenic lesion	200 mg/Kg PO	Dose-dependent complete inhibition of PLA2 ^b^ activation
Cervós-Navarro J. et al. [64]	1990	Comparative randomized study	Mongrel cats (2.5–4.5 Kg)	Ultraviolet-induced brain edema	100 mg/Kg IV (3 doses in 24 h)	Significant acceleration of the reabsorption of brain edema
Dixon C.E. et al. [69]	1997	Comparative study	Adult male Sprague Dawley rats (250–275 g)	Cortical impact injury	100 mg/Kg/18 d	Significant reductions on cognitive deficits and increase of extracellular acetylcholine levels
Plataras C. et al. [70]	2000	Comparative study	Albino Wistar rats	Incubation of homogenates of whole brain	0.1–1 mM	Stimulation of brain acetylcholinesterase and Na+/K+-ATPase
Başkaya M.K. at al. [71]	2000	Comparative study	Sprague Dawley rats (250–300 g)	Controlled cortical impact	50–400 mg/Kg IP 2 times after injury	Significant dose-dependent reduction of brain edema and blood−brain barrier disruption
Dempsey R.J. et al. [72]	2003	Comparative study	Adult male Sprague Dawley rats (250–280 g)	Controlled cortical impact	100–400 mg/Kg IP	Significant decrease of hippocampal neuronal death, cortical contusion volume, and neurological dysfunction
Menku A. et al. [73]	2010	Comparative study	Male Swiss albino rats (200–250 g)	Blunt trauma	250 mg/Kg IP	Significant reduction of Malonyldialdehyde levels with citicoline alone or in combination with propofol
Qian K. et al. [75]	2014	Comparative randomized study	Adult male Sprague Dawley rats (290–330 g)		250 mg/Kg IV 30 min and 4 h after	Marked reduction of brain edema and blood−brain barrier permeability, enhancement of the activities of superoxide dismutase and the levels of glutathione, reduction of the levels of malondialdehydeand lactic acid. Reduction of axonal damage and neuronal death
Schmidt K. et al. [67]	2015	Comparative randomized study	Male Wistar rats	Endotoxemia induced by Lipopolysaccharide injection	100 mg/Kg IV	Significant reduction of microvascular permeability
Hernekamp J.F. et al. [68]	2015	Comparative randomized study	Adult male Wistar rats (250–300 g)	Burn edema	100 mg/Kg IV	Significant reduction of macromolecular efflux and reduction of leukocyte activation
Jacotte-Simancas A. et al. [74]	2015	Comparative randomized study	Male Sprague Dawley albino rats (~250 g)	Controlled cortical impact injury	200 mg/Kg IP starting 4 h after surgery, and thereafter daily until completing five injections	Significant improvement of memory deficits and reduction of interhemispheric differences in the volume of the hippocampal formation
Gan D, et al., [76]	2020	Comparative randomized study	Zebrafish larvae	In vivo TBI zebrafish model	2.5 mg/mL for the drug incubation	Activation of microglia, reduction of neuronal apoptosis and promotion of neuronal proliferation

^a^ N/A = Not available; ^b^ PLA2 = Phospholipase A2.

Citicoline holds some biochemical, pharmacological, and pharmacokinetic characteristics to be a potentially useful drug for the management of TBI [81] and citicoline has an appropriate profile for the treatment of the different brain ischemia related disorders, having different neuroprotective and neurorestorative properties (Table 2) [82,83,84,85]. 

## 3. Clinical Experiences of Patients with Traumatic Brain Injuries

As it has been shown in the previous section about the experimental data showing that citicoline can induce significant positive effects in different experimental models of TBI, citicoline seems to be a suitable drug for the management of patients suffering TBI. Thus, it is possible to say that citicoline has a pleiotropic effect on several steps of the ischemic cascade involved in the development of the TBI [84].

Many years ago, the effect of citicoline stimulating the ascending reticular activating system at the brain stem level was postulated to explain the effects of the drug on the consciousness level [84]. As citicoline could be considered as a valid pharmacological treatment for TBI, many clinical studies have been performed over time to assess if the drug would have beneficial effects in the treatment of patients with TBI. 

There are early published clinical data showing that citicoline can lead to recovery from neurological clinical symptoms and a return to a conscious state with an excellent safety profile [86].

### 3.1. Clinical Studies on Mild, Complicated to Severe Head Injuries

Table 3 summarizes chronologically all clinical studies evaluating the effects of citicoline in the management of patients with mild, complicated to severe traumatic brain injury.

The first double-blind randomized and placebo-controlled clinical trial was presented in 1979 by Misbach at al. [87]. In this study, the authors concluded that the use of citicoline was associated with better recovery in patients with severe TBI.

In another double-blind study, performed by Ayuso et al. in 1979, it was demonstrated the effectiveness of citicoline to treat patients with memory disorders of an organic base, in that case induced by bilateral electroshock [88]. 

De la Herrán et al. [89], in an open study with the 32 patients with severe TBI among other types of brain injuries, concluded that the administration of citicoline accelerated normalization of the consciousness state. Similar results and conclusions were obtained in other double-blind studies performed by Espagno et al. [90] and by Carcasonne and LeTourneau [91], the last one performed in children population.

Richer and Cohadon [92] performed a randomized, double-blind and placebo-controlled trial in a sample of 60 patients with severe TBI. Citicoline was administered at a dose of 750 mg/d intravenously for 6 days, and then intramuscularly for 20 days more. At 60 days, the number of patients who had recovered consciousness was significantly greater in the group receiving citicoline. At 90 days, it was also found that the highest rate of recovery of motor deficits was associated with the treatment with citicoline. 

Lecuire and Duplay [93], in a double-blind study, compared the effects of citicoline (750 mg/d/10 d IV) to those of meclofenoxate (3 g/d/10 d IV) in a sample of 25 patients (14 patients treated with citicoline and 11 patients treated with meclofenoxate). Statistical analysis of the results demonstrated significant effects in the citicoline treated group regarding the resolution of consciousness disorders, EEG pattern, and functional recovery. Shortly after, the same authors confirmed these positive results in an open label study performed in a group of 154 patients with TBI injury [94]. Lecuire [95] conducted another double-blind study comparing piracetam (6 g/d) versus citicoline (750 mg/d) in a group of 40 patients with head injury. The results of the study showed a better result on consciousness status, vegetative and electric, and on the global final improvements in the group of patients treated with citicoline.

Cohadon et al. [96] demonstrated the clinical efficacy of citicoline in a double-blind placebo-controlled trial in a sample of 60 patients with severe TBI. A group of 30 patients was treated with citicoline (750 mg/intravenously for 6 days and continued up to 20 days more with intramuscular administration). In the treated group a shortening of the comatose period and an acceleration of the recovery of neurological deficits, especially in the motor area, were observed, these differences being statistically significant compared to placebo. The authors attributed these positive results to the effect of the drug on brain edema. Deleuze et al. [97] correlated the effectiveness of citicoline with its effect on the cerebral metabolism, reflected in a significant reduction of lactate levels in cerebrospinal fluid after the treatment.

Ogashiwa et al. [48] performed a study in a sample of patients with disturbances of consciousness associated with stroke, TBI or brain tumours. Fifty-one patients were treated with citicoline (1000 mg/d/7 d IV) and 50 patients with the same characteristics were used as controls. The effects were evaluated using the principal component analysis score and the global improvement rate. The results of the principal component analysis scoring correlated closely with those of the global improvement rate, the effects in the citicoline-treated group being significantly greater than those obtained in the control group. Citicoline was more effective on the items related to the performance than on the verbal factor. These authors considered the drug to be safe, and they even administered the drug by the intrathecal route in some cases [98,99].

In another controlled study, De Blas et al. [100] evaluated the effects of citicoline on the short- and long-term evolution in a group of 100 patients with head injuries, compared with a group of 100 patients treated conventionally. The results obtained suggested that the addition of citicoline to the conventional treatment regimen was associated with a decrease in the length of post-traumatic coma and the incidence of neurological and psychological sequelae, accelerating the recovery of these kind of deficits.

Raggueneau and Jarrige [101] published the results of a national inquiry conducted in 24 neurosurgical intensive care units in France. The authors obtained information on 921 cases of severe TBI. Among the total sample, 219 patients were treated with citicoline. This, then, allowed the comparison of the results obtained between patients treated and not treated with citicoline. The improvement of the outcome for all patients was significantly linked to citicoline treatment. Nevertheless, no effects on the mortality rate were seen associated with the use of the drug. 

Calatayud Maldonado et al. [102] conducted a single-blind randomized clinical trial in a sample of 216 patients with moderate to severe TBI with the objective of assessing the influence of the addition of citicoline to the standard treatment for head injury. One hundred and fifteen patients received treatment with citicoline (up to 4 g/d parenterally). The total duration of the treatment varied according to the evolution of the patient. The analysis of the results showed that citicoline significantly decreased hospital stay. Similarly, the treatment with citicoline was associated with a significant better global outcome, as evaluated with the Glasgow Outcome Scale, that was more relevant in the subgroup of patients with severe TBI. The duration of outpatient follow-up was also reduced in the group of patients treated with citicoline. 

Lozano [103] reported the results of a randomized study to assess the impact of the use of citicoline therapy on the evolution of patients with severe TBI. Citicoline was administered to 39 patients at a dose ranging from 3 to 6 g/d by intravenous infusion for 2 weeks. The results were compared with another group of patients with the same characteristics and not treated with citicoline. After 14 days of treatment, cerebral edema was significantly reduced or normalized in a higher number of patients treated with citicoline. Mean hospital stay was also significantly reduced in the active treatment group (28.718 ± 21.6 days) in comparison with control group (37.323 ± 35.22 days; *p* < 0.001). Regarding the final outcome, evaluated with the Glasgow Outcome Scale, it was a trend to have a better outcome in the group receiving the active treatment, but these differences did not reach statistical significance, probably due to the small sample size.

Lazowski et al. [104] performed a randomized and placebo-controlled study on a sample of 28 patients with traumatic brain injury caused by isolated head trauma. Citicoline was administered at a dose of 1 g IV for 14 days in addition to typical treatment. The GCS and the Glasgow Outcome Scale (GOS) were used to control patients up to 30 days. In the citicoline-treated group the analysis found no correlation between the GCS scores in day 7 and day 14, and this lack of correlation could be interpreted as a result of treatment with citicoline, and the significant correlation found on the GCS at 14 and 21 days could be interpreted as an expanded effect of treatment up to 21 days. In the citicoline-treated group, the GCS score at 21 days was significantly correlated with GOS scores at 30 days, showing the protective effect of the used drug.

Hinev al. [105] in their study observed that 80% of patients with severe head trauma recovered from neurological symptoms and unconsciousness, concluding that the use of citicoline was associated with reduced coma duration and accelerated recovery of neurological disturbances in patients with severe head trauma, highlighting the safety of the drug. 

Krishna et al. [106] conducted a randomized, single-blind, placebo-controlled, single-center, prospective trial in a sample of 100 patients. Patients were randomized to receive citicoline (2 g/d/60 d PO) or placebo and the evaluations of outcomes were made at discharge and after 30 and 90 days. The authors concluded that the rate of recovery was earlier in the citicoline group in terms of a shorter duration of stay, early gaining of full consciousness and relief from cognitive symptoms. 

The Citicoline Brain Injury Treatment Trial (COBRIT) was a double-blind randomized and placebo-controlled trial with a special design [107,108]. The objective of the trial was to determine the ability of citicoline to positively affect the functional and cognitive status in patients with complicated mild, moderate, and severe TBI. The primary outcome of the study was the functional and cognitive status at 90 days. The outcome was measured by the nine components of the TBI Clinical Trials Network Battery, that includes the Trail Making Test (parts A and B), the extended Glasgow Outcome Scale (GOS-E), the California Verbal Learning Test II, the Controlled Oral Word Association Test, some of the tests included in the Wechsler Adult Intelligence Scale III (Processing Speed Index and Digit Span), and the Stroop Test (Parts 1 and 2). The sample size was calculated assuming an odds ratio (OR) of 1.4 or higher, and the final sample size was fixed as 1296 patients, after adding 15% for presumed losses. The patients were randomized to receive either citicoline (2 g/d/90 d) or placebo by enteral route within 24 h after injury. The clinical trial was stopped early for futility with 1213 patients included. Rates of favorable improvement for the GOS-E were 35.4% in the citicoline group and 35.6% in the placebo group. For the other scales, the rate of improvement ranged from 37.3% to 86.5% in the citicoline group and from 42.7% to 84.0% in the placebo group. There were no significant differences between groups at the 90-day evaluation: global OR: 0.98 (95% CI: 0.83–1.15), nor at the 180-day evaluation: global OR 0.87 (95% CI: 0.72–1.04). On the basis of the results obtained, the authors concluded that citicoline, compared with placebo, was not effective in the improvement of the functional and cognitive status of patients with TBI.

Despite the COBRIT trial being the largest study performed with citicoline in the management of TBI, there are some methodological issues that could question the validity and applicability of the results obtained. This study was an independent and academic study, financed by the US National Institute of Health, with a somewhat limited budget. A first point to consider was the sample size calculation that was based on an assumption of an OR of 1.4 as the effect of the treatment; however, this assumption was arbitrary and not based on previous data. Then, it looked as though the sample size was calculated based on the budget rather than on previous data. With a more realistic OR of 1.2 or less the sample size would have been much higher and unaffordable for the authors. Another key point to consider was the inclusion of mixed populations, including mild, moderate, and severe TBI. The pathophysiology and the evolution can be largely different among these groups. To consider these differences, it is mandatory to use a randomized and matched sample design, which was not used in the COBRIT trial. Thus, this is an evident source of heterogeneity, but it has not been considered as an important confounding factor in the analysis and interpretation of the data. Another point to take into the account was the atypical oro-enteral administration of the drug that is not approved in any country and has not previously been tested in any way. The use of this route of administration is not recommended in patients with moderate or severe TBI, at least in the first days. However, the most controversial point was the poor compliance of the treatment. A compliance of only 44.4% of patients having taken more than 75% of the medication expected is exceptionally low and needs to be explained, explanation that was not included in the publication of the trial. We must consider that not receiving the active treatment is not the same as not receiving the placebo, in terms of the standard of care being received. A placebo is a substance or treatment which is designed to have no therapeutic value. In other words, less than half of the patients in the active drug group received something close to a therapeutic dose of citicoline. Thus, this makes it exceedingly difficult to assume a lack of effect of the drug when the patients did not receive the proper treatment regimen. 

El Reweny et al. [109] communicated the results of their study on patients with severe head injury. In their study 40 patients were allocated to 2 groups, where patients in Group I were treated with citicoline (1 g/d/14 d IV) in front of patients in Group II that received conventional therapy. They found that those patients in Group I with brain edema had the best outcome. Indeed, patients in Group II with intracerebral hematoma had the worst outcome. The authors concluded that the addition of citicoline to the conventional therapy of patients with severe TBI offered a trend to improve the outcome. Interesting are the results obtained by Varadaraju and Ananthakishan [110], demonstrating a certain synergistic effect when citicoline was administered together with cerebrolysin, as the patients treated with this association had a better outcome than patients treated with citicoline alone. Titov et al. [111] also demonstrated a positive effect of the combination of citicoline and cerebrolysin in the management of TBI in the acute phase. 

Trimmel et al. [112] investigated the potential role of citicoline administration in TBI patients treated at the Wiener Neustadt Hospital. In a retrospective subgroup analysis, they compared 67 patients at the study site treated with citicoline (3 g/d/3 weeks IV) and 67 matched patients from other Austrian centers not treated with citicoline. Patients with moderate to severe TBI were included. The analysis found a significant effect of citicoline, expressed by the reduction of the rates of mortality at the intensive care unit mortality (5% vs. 24%, *p* < 0.01), during the hospital stay (9% vs. 24%, *p* = 0.035), and after 6 months of follow up (13% vs. 28%, *p* = 0.031). A significant reduction in the rates of unfavorable outcome (34% vs. 57%, *p* = 0.015) was also detected and in the observed vs. expected ratio for mortality (0.42 vs. 0.84) in the citicoline group (Figure 2). Ahmadi et al. [113] published a double-blind, randomized clinical trial on 30 patients with severe TBI. According to the protocol (IRCT20140611018063N7) and the abstract, patients were randomly divided into three groups: A (control), B (citicoline 0.5 g/12 h/24 d IV), and C (citicoline 1.5 g/12 h/14 d IV), but once the authors explained the methods in the article, these groups changed to: A (citicoline 0.5 g/12 h/24 d IV), B (citicoline 1.5 g/12 h/14 d IV), and C (placebo). This incongruence makes it difficult to interpret the results, because if the group assignment was the original, then a significant dose-dependent effect of citicoline can be found, but with the assignment stated in the paper, the results are difficult to interpret, but the authors concluded that citicoline had no positive effect on the outcome of such patients.

In recent years, the roles of inflammation and oxidative stress have been highlighted as targets to neuroprotection in the management of TBI [10,11,12,13,14,15,16]. Thus, the measurement of the effects of these neuroprotective therapies on the levels of established biomarkers of inflammation and oxidative stress could be of interest to evaluate the efficacy of such treatments.

Salehpour et al. [114,115] assessed the effects of citicoline in a sample of patients with diffuse axonal injury in a double-blind and randomized clinical trial. The efficacy of citicoline was assessed by the measurement of malondialdehyde (MDA) levels in plasma as a marker of oxidative stress. The MDA levels at the different times of blood sampling improved significantly, whereas the control group showed no difference. The authors concluded that citicoline is an effective neuroprotective agent and can reduce MDA levels in patients with severe TBI and diffuse axonal injuries. 

Shokouhi et al. [116] conducted a double-blind and randomized clinical trial on 58 patients with the diagnosis of diffuse axonal injury and severe TBI to investigate the effects of citicoline on serum levels of fetuin-A and matrix Gla-protein (MGP), that are related with the inflammation and the vascular calcification secondary to head trauma. The findings suggested that citicoline may have protective effects against inflammatory damage and vascular calcification in TBI patients through increasing plasma levels of fetuin-A and MGP.

**Table 3 pharmaceuticals-14-00410-t003:** Summary of clinical studies evaluating the effects of citicoline in the management of patients with mild to severe traumatic brain injury.

Authors	Year	*n*	Severity	Type of Study	Control	Time Window	Schedule of Treatment	Follow-up	Main Results
Misbach et al. [87]	1978	80	Moderate to severe	Double blind RCT ^a^	Placebo	NA ^b^	300 mg/d/14 d IV	14 d	Better recovery rate (GCS ^c^)
Espagno et al. [90]	1979	46	Severe	Double blind RCT ^a^	Placebo	NA ^b^	250 mg/d/5 d IV + 250 mg/d/15 d IM	30 d	Better recovery of consciousness
Carcasonne & LeTourneau [91]	1979	43	Moderate to severe (children)	Double blind RCT ^a^	Placebo	NA ^b^	NA^b^	20 d	Faster recovery from coma
Richer & Cohadon [92]	1980	60	Severe	Double blind RCT ^a^	Placebo	24 h	750 mg/d IV (6 d) + IM (14 d)	90 d	More independent patients (clinical evaluation)
Lecuire & Duplay [93]	1982	25	Moderate to severe	Double blind RCT ^a^	Meclophenoxate	24 h	750 mg/d/10 d IV	10 d	More patients with a favorable outcome
Lecuire & Duplay [94]	1982	154	Moderate to severe	Open study	Bibliographic data	24 h	750 mg/d/10 d IV + 250 mg/d/10 d IM	20 d	Significant improvement of survival and resolution of neurological deficits and consciousness troubles
Cohadon et al. [96]	1982	60	Severe	Double blind RCT ^a^	Placebo	24 h	750 mg/d IV (6 d) + IM (20 d)	120 d	More independent patients (~GOS ^d^)
Lecuire [95]	1985	40	Moderate to severe	Double blind RCT ^a^	Piracetam	24 h	750 mg/d/10 d IV	10 d	Global result favorable to citicoline (*p* < 0.01)
Deleuze et al. [97]	1985	11	Severe	Open study	None	24 h	500 md IV single dose	4 d	Significant decrease of lactate and lactate/pyruvate ratio in CSF ^e^
De Blas et al. [100]	1986	100	Moderate to severe	Open RCT ^a^	Control	24 h	200–400 mg/8 h IV or IM in the acute phase, followed by 100–200 mg/8 h PO during follow-up	180 d	Reduction of coma and neurological and psychological sequelae
Ragguenneau & Jarrige [101]	1988	921	Severe	Cohort study	Control	24 h	500–750 mg/d/20 d IV	180 d	More independent patients (~GOS ^d^)
Calatayud Maldonado et al. [102]	1991	216	Moderate to severe	Single blind RCT ^a^	Control	24 h	3–4 g/d/4 d IV + 2 g/d/26 d PO	90 d	More independent patients (GOS ^d^) Decreased hospital stay
Lozano [103]	1991	78	Severe	Single blind RCT ^a^	Control	24 h	3–6 g/d/14 d IV	90 d	Trend to have more independent patients (GOS ^d^) Reduction of brain edema (CTscan) Decreased hospital stay
Lazowski et al. [104]	2003	28		RCT ^a^	Placebo	NA ^b^	NA ^b^	30 d	GCS 21 is significantly correlated with GOS 30 (r = 0.68; *p* < 0.01) showing the protective effect of citicoline
Hinev et al. [105]	2007	8	Severe	Open	None	36 h	1 g/d/5–7 d IV	NA ^b^	80% of patients recovered from neurological symptoms and un-consciousness
Krishna et al. [106]	2012	100	Moderate to severe	Single blind RCT ^a^	Placebo	24 h	2 g/d/60 d PO	90 d	Earlier rate of recovery, less duration of stay, early gaining of full consciousness and relief from cognitive symptoms
Zafonte et al. [108]	2012	1213	Mild, complicated, moderate and severe	Double blind RCT ^a^	Placebo	24 h	2 g/d/90 d PO or enteral	180 d	No differences on the TBI-Clinical Trials Network Core Battery
El Reweny et al. [109]	2012	40	Severe	Open RCT ^a^	Control	NA ^b^	1 g/d/14 d IV	NA ^b^	Trend to improve the outcome
Salehpour et al. [114]	2013	40	Severe with diffuse axonal injury	Single blind RCT ^a^	Control	24 h	2 g/d/12 d IV	12 d	Reduction of MDA plasma levels
Shokouhi et al. [116]	2014	58	Severe with diffuse axonal injury	Double blind RCT ^a^	Control	24 h	2 g/d/15 d IV	15 d	Increased plasma levels of fetuin-A and matrix Gla-protein
Salehpour et al. [115]	2015	40	Severe with diffuse axonal injury	Single blind RCT ^a^	Control	24 h	2 g/d/15 d IV	15 d	Reduction of MDA plasma levels No differences on GCS ^c^
Varadaraju et al. [110]	2017	60	Mild to moderate	Open RCT ^a^	Citicoline + Cerebrolysin	NA^b^	2 g stat followed by 500 mg IV/PO twice daily continued for 3 months.	180 d	The association had better outcome (GOS) than patients treated with citicoline alone
Trimmel et al. [112]	2018	134	Moderate to severe	Retrospective matched pair analysis	Control	24–48 h	3 g/d/21 d IV	180 d	Reduction of the rates of mortality Reduction of the rates of unfavorable outcome (GOS)
Ahmadi et al. [113]	2020	30	Severe	Double blind RCT ^a^	Control	NA ^b^	1–3 g/d/14 d IV	30 d	According to protocol: significant dose-dependent effect on outcome (GOS)
									According to article: no positive effect

^a^ RCT = randomized clinical trial; ^b^ NA = not available; ^c^ GCS = Glasgow Coma Scale; ^d^ GOS = Glasgow Outcome Scale; ^e^ CSF = cerebrospinal fluid.

### 3.2. Clinical Studies on Mild Head Injuries

Levin [117] conducted a pilot double-blind, randomized and placebo-controlled study with 14 patients with post-concussion syndrome associated with a mild TBI. Treatment with citicoline (1 g/d) for one month was associated with an improvement in memory tests, such as the Galveston Orientation and Amnesia Test, which was statistically significant as compared to placebo. The use of citicoline was linked to a higher symptomatic improvement, except for the gastrointestinal discomfort that was more frequent in the citicoline group, and for dizziness that was significantly more common in patients from the placebo group, at the end of the study. 

However, Aniruddha et al. [118], in a simple-blind, randomized and placebo-controlled study performed in a sample of 44 patients with mild head injury were unable to evidence differences between citicoline and placebo in relation to the evolution of the post-concussion symptoms. Despite that, citicoline could be considered a therapeutic option for post-concussion syndrome associated with mild TBI [119].

### 3.3. Clinical Studies on Cognitive Disorders Associated to TBI

León-Carrión et al. [120,121,122] focused their investigations on the effects of citicoline on memory disorders associated to TBI. These authors assessed the effects of the administration of a single dose of 1 g of citicoline on cerebral blood flow measured by the 133Xe inhalation technique in patients with severe memory deficit after TBI. Two measurements were made: baseline and at 48 h later. Patients received the drug one hour before the first test. In the first measurement, a significant hypoperfusion was detected at the inferoposterior area of the left temporal lobe, an area related with memory. This hypoperfusion disappeared after citicoline administration, showing an objective effect of citicoline normalizing the cerebral blood flow in the affected areas. In another study, these authors demonstrated that neuropsychological rehabilitation associated with citicoline achieved improvements in all the evaluated areas, especially in verbal fluency and the word recall Luria test, these differences being statistically significant when compared with placebo. Thus, citicoline can be considered as a valid pharmacological option for the management of cognitive disorders associated with TBI [123], and this effect also induces an improvement in the quality of life [124].

### 3.4. Meta-Analysis on the Effects of Citicoline in the Management of TBI

In 2014, a meta-analysis based on 12 clinical studies was published [125]. A systematic search of the relevant terms was performed to identify comparative clinical trials of citicoline in the acute phase of patients with mild, moderate, or severe head injuries. The primary efficacy measure was the rate of independence or good outcome at the end of a scheduled follow-up period. This meta-analysis involved a total of 2706 patient. The use of citicoline was associated with a significant increase in the rates of independence with an OR of 1.815 (95% CI: 1.3022.530), under the random effects model (Figure 3), and with an OR of 1.451 (95% CI: 1.224–1.721), under the fixed effects model.

In a more recent meta-analysis [126], the authors found neutral effects of citicoline in the treatment of patients with TBI, but this meta-analysis was based only on studies published in English, and that is a well-known source of bias, enough to question the results obtained. To clarify the role of citicoline in the management of TBI, a new meta-analysis is ongoing, and the results will be available soon (PROSPERO CRD42021238998).

## 4. Conclusions

Citicoline, also named cytidine 5′-diphosphocholine or CDP-choline, is an intermediate in the generation of phosphatidylcholine from choline in mammals, the formation of endogenous citicoline being one of the principal steps in the biosynthetic pathway of phosphatidylcholine [84].

Independently of the administration route, oral or parenteral, citicoline splits in its two principal components, cytidine and choline. The absorption by the oral route is almost complete, and the bioavailability by the oral route is approximately the same as the intravenous route. Citicoline is widely distributed all around the body, crosses the blood−brain barrier and reaches the central nervous system (CNS). In the CNS, citicoline is incorporated into the phospholipid fraction of the cells, mainly at membranal and microsomal level, especially in neurons. Citicoline increases the biosynthesis of phosphatidylcholine in neuronal membranes, improves the brain energetic metabolism, and can modulate the levels of different neurotransmitters, such as acetylcholine, dopamine, and norepinephrine. Due to these biochemical and pharmacological actions, citicoline has demonstrated neuroprotective effects in hypoxic and ischemic conditions of the brain. As described in this review, citicoline can restore the activity of mitochondrial ATPase and membrane Na+/K+-ATPase, and it is able to normalize the activity of phospholipase A2, and these actions leads to an acceleration of the reabsorption of cerebral edema in various experimental models. Thanks to its action as a choline donor, increasing the levels of acetylcholine, citicoline can improve the learning and memory performance in some animal models of brain aging [84]. 

In this review, it has been shown that in patients with moderate to severe TBI, the addition of citicoline into their standard therapeutic regimen could offer some benefits, as this drug can accelerate cerebral edema reabsorption and recovery, resulting in a shorter hospital stay and improved final outcome, with a higher independence rate among the patients treated with citicoline. However, despite the progressive increase of the doses, the beneficial effect of citicoline over time has been diluted in parallel with the improvement of the standards of care of TBI [125].

Citicoline is a safe drug, as shown by the toxicological studies conducted, without a significant systemic cholinergic effect and is a well-tolerated product as it has been shown in the Periodic Safety Reports made since its commercialization in the 1970s. Even in a Cochrane review for the evaluation of the effect of citicoline on elderly people with behavioral and cognitive disturbances [127], a lower rate of incidence of adverse events related with citicoline was demonstrated, in comparison with placebo.

The pharmacological and biochemical properties and the complex mechanisms of the action of citicoline suggest that this product is indicated for the management of acute stroke, both ischemic and hemorrhagic, TBI of varying severity, and cognitive disorders of different origins. The available data obtained from clinical studies in the management of patients with TBI indicate that citicoline can accelerate cerebral edema reabsorption and recovery, resulting in a shorter hospital stay and improved final outcome, with a higher independence rate among the patients treated with citicoline. These effects could be explained by the pharmacodynamics of the product and its pleiotropic effect on the mechanisms involved in the development of the TBI [41,84], some of these mechanisms are considered of interest as targets for the development of neuroprotective strategies for the management of TBI [128,129]. Citicoline also was able to induce an improvement of the cognitive disorders associated with TBI. No serious side effects have occurred in any series of patients treated with citicoline [41,84,130].

Depending on the results of the new meta-analysis (PROSPERO CRD42021238998), it could be valuable to plan further clinical studies, specially focused on severe TBI and with higher doses of citicoline. However, no new large clinicals trials with citicoline in this field are expected.

## Figures and Tables

**Figure 1 pharmaceuticals-14-00410-f001:**
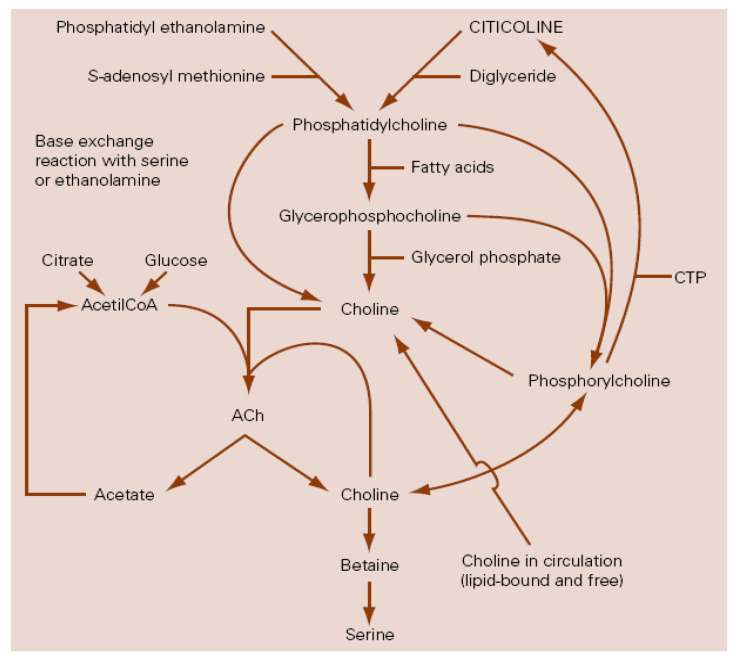
Relationship between citicoline and choline metabolism, cerebral phospholipids and acetylcholine. ACh: Acetylcholine; CTP: Cytidin triphosphate (Adapted with permission from Secades, J.J.; Lozano, R. Traumatismos craneoencefálicos: revisión fisiopatológica y terapéutica. Aportaciones de la citicolina. Excerpta Medica. Amsterdam, The Netherlands, 1991. Copyright 1985 Ferrer Internacional S.A.).

**Figure 2 pharmaceuticals-14-00410-f002:**
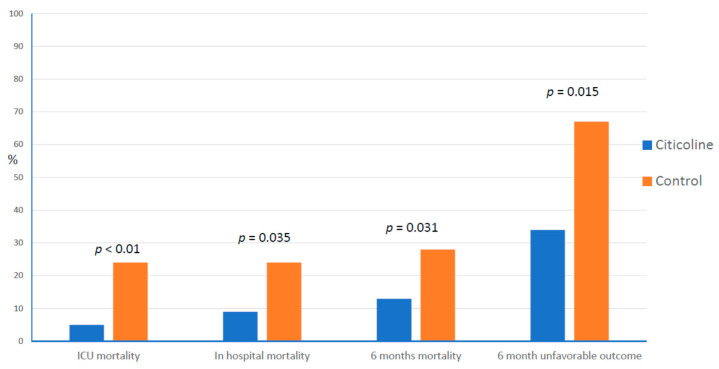
Effect of citicoline treatment on the rates of mortality and unfavorable outcome. ICU = Intensive Care Unit.

**Figure 3 pharmaceuticals-14-00410-f003:**
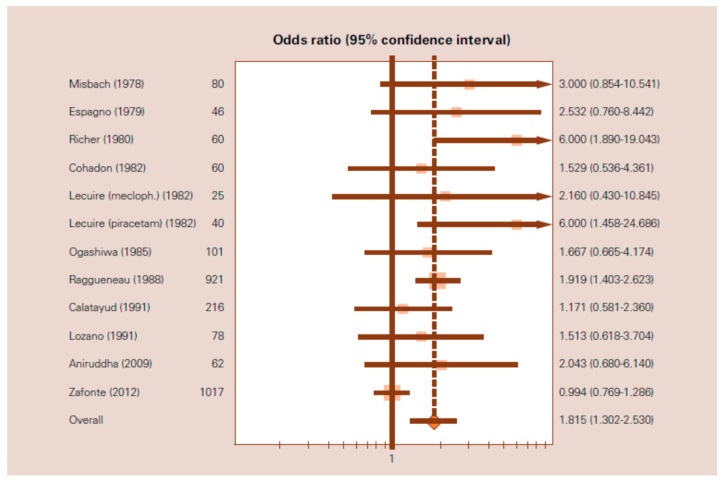
Forest plot of the meta-analysis of the effects of citicoline on independence after traumatic brain injury. OR = 1.815 (95% CI:1.302–2.530) based on the random-effects model. (From Secades, J.J. Citicoline for the Treatment of Head Injury: A Systematic Review and Meta-analysis of Controlled Clinical Trials. *J. Trauma. Treat*. **2014**, *4*, 227).

**Table 2 pharmaceuticals-14-00410-t002:** Main actions of citicoline on experimental models of TBI.

Protection and restoration of neuronal membrane. Normalization of phospholipid content in membranes. Normalization of neuronal membrane functions. Normalization of ionic exchange across neuronal membrane. Restoration of some enzymatic activities (CTP:phosphocholine cytidyltransferase, …). Improvement of neurotransmission (acetylcholine, dopamine, etc.). Improvement of cerebral metabolism. Restoration of the activity of membrane-bound ATPases. Inhibition of the activity of phospholipases, avoiding the release of free radicals, and second messengers. Empowerment of antioxidative and anti-inflammatory mechanisms. Acceleration of the reabsorption of brain edema. Reduction of the volume of brain ischemic lesions. Inhibition of apoptosis. Activation of cell repair mechanisms and neuroplasticity.

## Data Availability

No new data were created or analyzed in this study. Data sharing is not applicable to this article.

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
