# Peer review of "Role of Citicoline in the Management of Traumatic Brain Injury"

_pharmaceuticals, 2021, doi:10.3390/ph14050410_

Round 1
Reviewer 1 Report
This manuscript is a review of preclinical and clinical studies on citicoline in TBI. The concept and the broad message of the manuscript are worthy. The critique of COBRIT, which is largely responsible for the TBI community having abandoned citicoline, is appropriate. The brief review of two meta-analyses and the promise of a third one is appropriate.
Critique
Unfortunately, the broad message is lost in the poor presentation and poor organization of the manuscript. Major reorganization and rewriting would greatly strengthen the manuscript and thereby strengthen the message.
The experimental section is poorly organized and needs to be rewritten. One thing that would help is to organize according to outcome measured. Also, it is important that the model used be indicated, and that in vitro experiments be described separately from in vivo experiments. Also, the time to treatment – the treatment window – is critically important for translation yet is completely absent from this brief review.
Fig. 1 starts with ischemia, but the manuscript is about TBI. Please fix
The clinical section is poorly organized. It would be more useful if studies were grouped according to outcome. The standard outcome – clinical outcome at 3-6 moths, as measured by GOS, GOSE etc, -- should form one of the outcome blocks, the most important one, with its own subheading. Others, with their own subheadings, could be organized around arousal from coma, memory, decreased hospital stay, etc. As is currently, with everything mixed up, there is no clear discernable message.
Author Response
First of all, thanks for your comments
You recommend:
- Unfortunately, the broad message is lost in the poor presentation and poor organization of the manuscript. Major reorganization and rewriting would greatly strengthen the manuscript and thereby strengthen the message.
I tried to reorganize it, but also I maintain somehow a chronological narrative
- The experimental section is poorly organized and needs to be rewritten. One thing that would help is to organize according to outcome measured. Also, it is important that the model used be indicated, and that in vitro experiments be described separately from in vivo experiments. Also, the time to treatment – the treatment window – is critically important for translation yet is completely absent from this brief review.
I tried to improve it and added a table with the experimental data
- 1 starts with ischemia, but the manuscript is about TBI. Please fix
This figure has been eliminated as figure 2
- The clinical section is poorly organized. It would be more useful if studies were grouped according to outcome. The standard outcome – clinical outcome at 3-6 moths, as measured by GOS, GOSE etc, -- should form one of the outcome blocks, the most important one, with its own subheading. Others, with their own subheadings, could be organized around arousal from coma, memory, decreased hospital stay, etc. As is currently, with everything mixed up, there is no clear discernable message.
Section reordered according the different type of studies, according severity, sequelae and meta-analysis. Table included
Hope the changes I did would change your opinion about the review and then you can recommend its publication
Sincerely,
Julio J Secades MD, PhD
Reviewer 2 Report
The author presents a review article on the use of citicoline in the treatment of traumatic brain injury (TBI). The scope of the manuscript as presented extends to an overview of the pathophysiology underlying TBI and the multiple neuroprotective actions displayed by citicoline in the onset and evolution of TBI. The topic is interesting and popular. The manuscript was perfectly prepared/polished. Still, I have several comments and/or suggestions on how to improve this manuscript, at least in my view.
- The pathway of synthesis and metabolism of melatonin should be introduced briefly
- While citicoline is a well-tolerated drug, no systematic analysis of dose-dependency is presented from the studies, and no lower boundary described for its effects. It is better to summarize the results of experimental studies using tables (including dose, model, effect and mechanism, etc) to improve the clarity of the text.
- Quality of figure 4 must be improved.
- Please provide recommendation for future studies in the discussion section.
Author Response
First of all, thanks for your comments
You recommend:
- The pathway of synthesis and metabolism of melatonin should be introduced briefly
New figure introduced
- While citicoline is a well-tolerated drug, no systematic analysis of dose-dependency is presented from the studies, and no lower boundary described for its effects. It is better to summarize the results of experimental studies using tables (including dose, model, effect and mechanism, etc) to improve the clarity of the text.
Tables included and more data about safety are included too
- Quality of figure 4 must be improved.
Done
- Please provide recommendation for future studies in the discussion section.
- Done
Hope the changes I did would change your opinion about the review and then you can recommend its publication
Sincerely,
Julio J Secades MD, PhD
Round 2
Reviewer 1 Report
This version is improved. My concerns have been addressed. I have no further suggestions
PS: I look forward to the upcoming meta-analysis